# Analysis of Association between Breastfeeding and Vegetable or Fruit Intake in Later Childhood in a Population-Based Observational Study

**DOI:** 10.3390/ijerph17113755

**Published:** 2020-05-26

**Authors:** Jadwiga Hamulka, Monika A. Zielinska, Marta Jeruszka-Bielak, Magdalena Górnicka, Dominika Głąbska, Dominika Guzek, Monika Hoffmann, Krystyna Gutkowska

**Affiliations:** 1Department of Human Nutrition, Institute of Human Nutrition Sciences, Warsaw University of Life Sciences (SGGW-WULS), 02-787 Warsaw, Poland; monika_zielinska@sggw.pl (M.A.Z.); marta_jeruszka_bielak@sggw.pl (M.J.-B.); magdalena_gornicka@sggw.pl (M.G.); 2Department of Dietetics, Institute of Human Nutrition Sciences, Warsaw University of Life Sciences (SGGW-WULS), 02-787 Warsaw, Poland; dominika_glabska@sggw.pl; 3Department of Food Market and Consumer Research, Institute of Human Nutrition Sciences, Warsaw University of Life Sciences (SGGW-WULS), 02-787 Warsaw, Poland; dominika_guzek@sggw.pl (D.G.); krystyna_gutkowska@sggw.pl (K.G.); 4Department of Functional and Organic Food, Institute of Human Nutrition Sciences, Warsaw University of Life Sciences (SGGW-WULS), 02-787 Warsaw, Poland; monika_hoffmann@sggw.pl

**Keywords:** breastfeeding, school children, gender, vegetables, fruit, meals, social factors

## Abstract

Vegetable and fruit consumption in childhood remains below recommendations in many countries. As the intake of fruit and/or vegetables during childhood in a few studies was associated with breastfeeding, it may be suggested as a serious interfering factor while analyzing the association between breastfeeding and its long-term health effects. Thus, it may be important for creation and implementation of effective public health programs. The aim of this study was to evaluate the association between breastfeeding and consumption of fruit or vegetables in later childhood. The study was conducted within the Project “ABC of Healthy Eating—ABC of Kids and Parents” among a representative sample of 703 pairs of mothers and children aged 7–12 years in Poland. A systematic purposive-quota selection according to gender, age, place of residence, and region of Poland was applied. The study was conducted by interviewers in the respondents’ homes in 2017. Logistic regression analysis was conducted, and models adjusted for children’s age, BMI centile and gender, maternal education, employment status, and economic situation, as well as for EU–28 average Gross Domestic Product (GDP) region. Almost 86% of children were ever breastfed. Vegetables for breakfast, second breakfast, dinner, supper, and between meals were consumed by 23.9%, 18.6%, 47.4%, 26.7%, and 4.0% of children, whereas fruit was consumed by 13.9%, 46.1%, 7.7%, 12.9%, and 59.7% of children, respectively. Breastfeeding for a period of 4–6 months increased the chance of vegetable consumption for breakfast in the whole group (aOR 3.80, 95% CI 1.90–7.59, *p* ≤ 0.001) and particularly in girls (aOR 4.60, 95% CI 1.43–14.75, *p* ≤ 0.01) when compared to boys (aOR 3.17, 95% CI 1.32–7.63, *p* ≤ 0.01). Longer duration of breastfeeding (over 12 months) increased the chance of vegetable consumption for dinner in the total group (aOR 2.36, 95% CI 1.30–4.26, *p* ≤ 0.01) and particularly in girls (aOR 3.04, 95% CI 1.24–7.46, *p* ≤ 0.01) when compared to boys (aOR 2.20, 95% CI 1.01–4.95, *p* ≤ 0.05). We showed a positive association between breastfeeding and vegetable consumption for breakfast and dinner among children aged 7–12 years. These associations were gender-specific (stronger among girls) and were not diminished by socio-demographic factors.

## 1. Introduction

Taking into account the need for improvement of the nutritional status, growth and development, health, and thus the survival of infants and young children, the World Health Organization (WHO), in cooperation with United Nations Children’s Fund (UNICEF), developed the global strategy for infant and young child feeding [1]. Within the proposed comprehensive strategy, breastfeeding was indicated as one of the most important elements [1], which is in agreement with the Innocenti Declaration on the Protection, Promotion, and Support of Breastfeeding by UNICEF [2]. It results from the fact that during the past decades evidence for the health advantages of breastfeeding has continued to increase, so a specific recommendation to exclusively breastfed for the first six months has been promoted [3]. It is emphasized that breastfeeding saves more lives than any other preventive intervention and that optimal breastfeeding could each year save the lives of over 820,000 children under the age of 5 years [4]. It was confirmed by the results of meta-analysis by Sankar et al. [5], as they proved that children aged 6–11 and 12–23 months who were not breastfed had 1.8-fold and 2.0-fold respectively higher mortality risk, compared to those who were breastfed.

However, not only short-term effects but also potential long-term effects of breastfeeding are currently listed, and it is stated that it may have a broad effect on the future health in childhood, adolescence, and adulthood. In the recent systematic review and meta-analysis by WHO [6], it was specified that breastfeeding was associated with a 24% reduction in overweight and/or obesity, as well as that it was associated with an increase in 3.5 points in normalized intelligence test scores. Similarly, in the meta-analysis by Victora et al. [7], it was indicated that breastfeeding may protect against child infections and malocclusion, increase intelligence, and reduce the risk of excessive body mass and diabetes. It was confirmed by other meta-analysis for excessive body mass [8,9], intelligence [10], and diabetes [8].

However, in a number of systematic reviews and meta-analysis presenting long-term benefits associated with breastfeeding, it is emphasized that confounding cannot be ruled out, and it may be an important problem, which is also indicated by WHO [6]. Especially in the systematic review and meta-analysis by Weng et al. [11], not only breastfeeding but also other factors during infancy were indicated as those associated with the childhood overweight. Taking this into account, there is a need to define the associations between breastfeeding and other factors that may confound the observed associations with life-long health effects of breastfeeding. Especially for factors such as dietary patterns and physical activity patterns, which may be also significantly associated with non-communicable diseases and issues such as overweight/obesity, diabetes, and intelligence, the potential confounding is always a consideration [12].

In the American study by Perrine et al. [13], which was based on the Infant Feeding Practices Study II and Year 6 Follow-Up, it was stated that breastfeeding is associated with a number of healthier dietary patterns at age 6, including intake of water, fruit, and vegetables, while for intake of sugar-sweetened beverages and 100% juice, there was an inverse association. At the same time, in the Brazilian study by Soldateli et al. [14], conducted using data from a former randomized clinical trial of adolescent mothers, their children, and maternal grandmothers, it was stated that breastfeeding was positively associated with vegetable intake in children aged 4–7 years but not with the fruit intake. Similarly, in the Danish cohort study by Specht et al. [15], based on data from the Healthy Start primary intervention study, the Danish Medical Birth registry, and the Danish Health Visitor’s Child Health Database, it was observed that breastfeeding was associated with higher vegetable intake in children aged 2–6 years, but not with the fruit intake. As the intake of fruit and/or vegetables during childhood was associated in a few studies with breastfeeding, it may be suggested as a serious interfering factor while analyzing the association between breastfeeding and long-term health effects. Besides, many covariates may be related to breastfeeding and healthier eating patterns such as maternal education, marital status, parity, poverty/income ratio, etc. [13,14]. Taking it to account, it is necessary to verify this association also in other countries to indicate if the association between breastfeeding and intake of fruit and/or vegetables is typical or not. The aim of the presented study was to analyze the association between breastfeeding and fruit or vegetable consumption behaviors in the population-based representative sample of Polish children aged 7–12 years.

## 2. Materials and Methods

### 2.1. Ethics Statement

The project followed the ethical standards recognized by the Declaration of Helsinki and was approved by the Bioethics Committee of the Faculty of Medical Sciences, University of Warmia and Mazury in Olsztyn on 17 June 2010 (Resolution No. 20/2010). Parental or legal guardians’ written informed consent to participate was obtained. 

### 2.2. Study Design and Sample Collection 

The study was conducted as a cross-sectional quantitative survey under the 3rd edition of national multicenter Project “ABC of Healthy Eating—ABC of Kids and Parents”. According to the study design, recruitment and data collection were conducted by a research agency—GfK Polonia (Warsaw, Poland). Participants were selected from the total population of Polish children aged 7–12 years and their mothers or legal guardians. The study group was chosen in a systematic selection process by date from PESEL database (Universal Electronic System of Population Register; version from October 2013), that is, a personal identification number given to every Polish citizen. Quota selection using gender, age (based Central Statistical Office in Poland (CSO-GUS) data from 31.12.2016), place of residence, and region was used to ensure the representativeness of the Polish child population. The six distinct macroregions (south, north, east, south-west, central, and north-east) were captured. The sample was drawn in proportion to the distribution of the macroregion and size of the town for the population of all children aged 7–12 in Poland, which covered the primary school age at that time (grades 1st to 6th). To the quota drawn in this way, numbers divided into age and gender were applied. At this stage, the test sample was divided into 100 sub-groups, with 16 interviews (8 for children and 8 for mothers) to be carried out. Ultimately, 1600 interviews (*n* = 800 children, *n* = 800 mothers) were carried out. The study was conducted in August 2017 by professional interviewers at the respondents’ homes using computer-assisted personal interviewing (CAPI) technique. Face-to-face structured interviews in home visits with the children and their mothers were conducted. Final analysis included 703 mother-child pairs due to either lack or unreliable data (Figure 1).

### 2.3. Dietary Data

The interview with the child, in the presence of the mother, was conducted using the questionnaire. Children were asked about the frequency of meals (breakfast, second breakfast, lunch, dinner, supper) and the occurrence of selected food groups, i.e. vegetables and fruit in particular meals. Information about the frequency of consumption of all kinds of vegetables, excluding potatoes, and fruit were collected according to the following categories: (1) never or almost never, (2) less often than once a week, (3) once a week, (4) several times a week, (5) once a day, (6) several times a day according to a validated children’s questionnaire (acronym: SF-FFQ4PolishChildren) [16]. Additionally, data on snacking between meals were collected, namely, how often, what, and in which situations children ate the snacks. 

Next, the children’s answers were verified by their mothers. Mothers were also asked about any breastfeeding occurrence and its duration (in months or weeks).

### 2.4. Anthropometric Data 

Data about the weight and height of children were proxy-reported by the mothers. Body mass index (BMI, kg/m^2^) was calculated, and centiles of the BMI were calculated using the online OLAF calculator [17] based on Polish growth charts [18]. 

### 2.5. Socio-Demographic Data

Detailed sociodemographic data were collected:

Number of children younger than 18 years old in the household.Maternal education: (1) primary, (2) vocational, (3) high school, (4) university.Maternal employment status: (1) unemployed, (2) unearned income, (3) part-time job, (4) full-time job.Self-declared economic situation of household: (1) we live very poorly—we do have enough resources just for the cheapest food; (2) live very poorly—we do have enough resources just for the cheapest food, clothing, and housing fees; (3) we live modestly—we have enough resources for food, clothing, housing fees, and instalment loan; (4) we live relatively thriftily, but we do not save for the future; (5) we live very well—we can afford everything without limitations.Place of residence: (1) village, (2) town < 20,000 inhabitants, (3) town 20,000–50,000 inhabitants, (4) town 50,000-100,000 inhabitants, (5) city 100,000–200,000 inhabitants, (6) city 200,000–500,000 inhabitants, (7) >500,000 inhabitants.Place of residence (region): Voivodships of Poland.

For further analysis selected data were re-categorized and final categories were created:

For maternal employment status: non-working (combined two categories (1) unemployed and (2) unearned income) vs. working (combined two categories (3) part-time job and (4) full-time job).For self-declared economic situation of household: we live thriftily or poorly (combined four categories 1-4) vs. we live very well (only category no. 5).For place of residence (living area): rural (only category no. 1) vs. urban (combined six categories from 2 to 7).Three macroregions according to the gross domestic product (GDP), expressed as a percentage of the EU–28 average which is set equal to 100%, were extracted based on the Eurostat data: 47%–50% (voivodships: świętokrzyskie, podlaskie, lubelskie, podkarpackie, warmińsko-mazurskie), 51%–100% (voivodships: dolnośląskie, kujawsko-pomorskie, lubuskie, łódzkie, małopolskie, opolskie, pomorskie, śląskie, wielkopolskie, zachodniopomorskie), 101%–110% (voivodship: mazowieckie ) [19].

### 2.6. Statistical Analysis

Categorical variables were presented as a sample percentage (%). The differences between groups were verified by a Chi-square test for categorical data or, due to non-parametric distribution (confirmed by using Shapiro–Wilk test), for continuous data by U Mann–Whitney test. The associations between breastfeeding duration and vegetable and fruit consumption and variables under study were verified with a logistic regression. Analysis were carried out for total population, separately for boys and girls, for vegetable or fruit consumption separately for different meals (breakfast; second breakfast; dinner; supper) and between meals, as well as for number of times of vegetable and fruit consumption per day (0 times; ≥ 5 times). We created one univariate model (Model 1) and two multivariate models. Model 2 was adjusted for age (years), gender (in the total group), children BMI centile, maternal education, maternal employment status, and family economic situation, whereas Model 3 included covariates used in Model 2 and additionally macroeconomic region of Poland based on EU–28 average. The odds ratios (ORs) and 95% confidence intervals (95% CI) were calculated. For all tests, the two-sided significance level *p* ≤ 0.05 was considered as significant. Analysis were performed using Statistica 13.3 software (TIBCO Software Inc., Palo Alto, CA, USA). 

## 3. Results

### 3.1. Study Group Characteristics 

Among 703 mother-children pairs slightly dominated boys (53% vs. 47%) and children aged 7–9 years compared to 10–12 years (53% vs. 47% —Table 1). In the study group, 14.5% of children were never breastfed, whereas 12.5% were breasted for over 12 months. Over 70% of children had normal body weight while 12% were overweight, and an additional 8% were obese. Around half of the women were mothers of two children (48%) and had high school education (48%), and around two-third worked on full-time (64%). More than half of the families lived in urban areas (56%) and in macroregions with GDP at the level 51%–100% of EU-28 average, whereas more than 90% declared that they live thriftily or poorly. Considering the examined factors, no statistically significant differences were found depending on the age or sex of the respondents.

### 3.2. Vegetable and Fruit Consumption

Above 90% of children aged 7–12 years always consumed breakfast, dinner and supper, while only 42% consumed second breakfast (Table 2). Neither age group nor sex influenced the frequency of meal consumption. Children most often consumed vegetables for dinner (47% of children) and supper (27%), while fruit for second breakfast (46%) and between meals (60%) as snacks. There were no significant differences between both age groups in the consumption of fruit and vegetables for the particular meals; however, a higher percentage of older children than the younger ones ate vegetables for second breakfast (22% vs. 16%, *p* ≤ 0.05). On the contrary, a higher percentage of younger boys than the older ones consumed fruit for supper (15% vs. 8%; *p* ≤ 0.05). The differences between girls and boys were only found for the consumption of fruit between meals, and only in the older age group, where a higher percentage of girls than boys ate fruit as snack during a day (68% vs. 55%; *p* ≤ 0.01). Vegetables and fruit were consumed at least 5 times per day only by 11% of children, while 7% of children did not eat vegetables or fruit even once a day. No differences between never or ever breastfed children in vegetable or fruit consumption for particular meals were observed. However, among ever breastfeed children, a lower percentage of children that did not consume any vegetables per day were found (22.2% vs. 42.6%, *p* ≤ 0.05; data not shown).

### 3.3. Breastfeeding and Vegetable and Fruit Consumption

Table 3, Table 4, Table 5 and Table 6 present results of logistic regression models investigating the associations between breastfeeding and vegetable and fruit consumption. Compared to never breastfed children, those who were breastfed for 0–3 or 4–6 months had a higher odds of vegetable but not fruit consumption for breakfast after adjustment for children, maternal, and family confounders (Model 3): aOR 2.72 (95% CI 1.34–5.50), *p* ≤ 0.01 and aOR 3.80 (95% CI 1.90–7.59), *p* ≤ 0.001 (Table 3). That effect was also observed in girls: aOR 4.09 (95% CI 1.26–13.22), *p* ≤ 0.05 and aOR 4.60 (95% CI 1.43–14.75), *p* ≤ 0.01, respectively, for breastfed for 0–3 months and 4–6 months; as well as in boys but only breastfed for 4–6 months: aOR 3.17 (95% CI 1.32–7.63), *p* ≤ 0.01. For dinner, breastfeeding for at least 7 months increased around two times the odds of consuming vegetables in total group (Table 4). Those odds were higher among girls: aOR 2.61 (95% CI 1.22–5.58), *p* ≤ 0.01 and aOR 3.04 (95% CI 1.24–7.46), *p* ≤ 0.01, respectively, for breastfed for 7–12 months and over 12 months. Among boys, the significant differences were found only for at least 12 months breastfeeding duration (aOR 2.20 (95% CI 1.01–4.95), *p* ≤ 0.05). In Table 5, data for supper are presented. Girls breastfed for at least 12 months had 31% higher odds of consuming vegetables for supper, but no other associations were observed. For fruit, no relationships were detected for any of the abovementioned meals. Additionally, breastfeeding was neither related to the lack of vegetable and fruit in the diet (consumption 0 times per day) nor to the consumption that met dietary guidelines (at least 5 times per day) (Table 6). 

There were no associations between breastfeeding and vegetable consumption for the second breakfast (Appendix A). Girls who were breastfed for 0–3 months had twice higher odds of consuming fruit for the second breakfast: aOR 2.23 (95% CI 1.05–4.76), *p* ≤ 0.05. Breastfeeding was no related to vegetable or fruit consumption between meals (Appendix A). 

## 4. Discussion

The present study showed a positive association between breastfeeding and vegetable consumption for breakfast and dinner among children aged 7–12 years. These associations were gender-specific (stronger among girls) and were not diminished by socio-demographic factors. Children who were ever breastfed had significantly higher incidence of eating vegetables for breakfast and for dinner when compared to those never breastfed. This was specifically pronounced in girls. Interestingly, in a case of dinner during which vegetables were consumed by the highest percentage of children, an increasing tendency was observed in relation to breastfeeding, namely, longer duration of breastfeeding resulted in higher odds of vegetable intake in the total population, as well as in both girls and boys, also when the potential confounders were considered. However, there was no association between breastfeeding and total daily number of servings of vegetables and fruit. For fruit, no relationships with breastfeeding was observed, regardless of the children’s gender. 

Most of the studies that focused on association between breastfeeding (duration or exclusive breastfeeding) and later vegetable and/or fruit intake confirm such relationship, in some cases only for vegetables [14,15,20] or for both vegetables and fruit [13,21,22]. A positive association between duration of breastfeeding and greater consumption of vegetables, but not fruit, was found among Brazilian children aged 4–7 years old [14] and in Danish children aged 2–6 years [15]. Burnier et al. [23] concluded that longer exclusive breastfeeding was a predictive factor for higher vegetable consumption in Canadian toddlers. Cooked vegetables, uncooked vegetables, and fruit were consumed more often at 7 years of age, among children breastfed for over 16 weeks when compared to non-breastfed children [21]. In 4 European cohorts of young children, longer breastfeeding duration was related to higher vegetable intake and also to fruit intake, but only in French (EDEN) and in British (ALSPAC) cohorts, while for the remaining two cohorts (Portuguese Generation XXI and Greek EuroPrevall), the results were less consistent [24]. The differences between countries (and across the countries) were also indicated by De Wild et al. [25]. 

There might be a few possible explanations of the positive association between breastfeeding and vegetable or fruit consumption. First, infants who are breastfed (ever or for longer period) have higher exposition to a greater variety of flavors that are transmitted from the mother’s diet via breast milk, which may increase the acceptance of novel foods, specifically vegetables, during the weaning period and in later life [26]. Moreover, repeated exposure to a variety of flavors via breast milk may also increase the children’s preferences of vegetables (and fruit), which further results in higher intake of those foods [27,28]. Secondly, mothers that breastfeed their child might have higher health consciousness, and thus offer vegetables and fruit to their children more often and a healthier diet in general [29]. These women may also have healthier eating habits, including vegetable and fruit intake [30]; therefore, they may create favorable conditions and serve as a good example to their child [14]. Nevertheless, higher fruit and particularly vegetable intake in young children related to longer breastfeeding duration was not explained by maternal fruit and vegetable intake in a study across 4 European cohorts [24]. Finally, breastfeeding promotes parenting feeding styles that are less controlling and more responsive to infant reactions, which increases infant responsibility for food intake. This, in turn, may result in less picky eating among children in later life. Food neophobia and pickiness negatively influence vegetable and fruit intake [23,31,32]. However, in our study, we found an association between breastfeeding and vegetable consumption on main meals like breakfast and dinner but no significant association with the total number of servings of vegetables and fruit during a day. Such specific relationships should be addressed in further research when also other methods of dietary data collection would be applied.

The lack of association between breastfeeding and fruit intake found in our study and in some others might be due to different sensory properties of vegetables vs. fruit. Sweet flavor, more typical of fruit than of vegetables, is the most preferred flavor in humans, which is inborn as sweetness is related to readily available calories from carbohydrates [26,33]. On the contrary, inborn rejection of bitter flavors, which are characteristic of plenty of vegetables, has to be overcome by repeated experience with such flavors [28,33]. This, as was mentioned above, takes place during breastfeeding as long as the vegetables are part of the mother’s diet [26]. Another explanation of why breastfeeding was not related to fruit intake in our study and in other ones may be that fruit is the most common first complementary food given to infants in many countries. In turn, this is associated with generally high fruit intake among children, and as such, the differences, particularly significant ones, are more difficult to detect [20].

Our study showed that the association between breastfeeding and vegetable intake was more pronounced among girls than boys. One of the reasons might be that among girls, higher heterogeneity in consumption of vegetable (combined with fruit) was observed. Among girls in the older age group, the numbers ranged from 4.6% (neither vegetables nor fruit were eaten during or between meals) to 32.2% (when vegetables and fruit were eaten 3 times per day), while for boys the numbers ranged from 7.9% to 32.0% (when vegetables and fruit were eaten 2 times per day). Besides, among girls, a shift into higher frequencies of vegetable and fruit consumption was found. Higher or more frequent intake of fruit and vegetables among girls than boys is known from the literature [34,35]. 

The association between breastfeeding and vegetable and/or fruit consumption can be modified by potential confounders. According to the literature, maternal education and socio-economic status are among the strongest ones. Children of more educated mothers tend to have higher intake of vegetables [23,36] and diets of higher quality in general [22,37,38]. Similar direct association between socio-economic status of mothers and eating habits of their children is observed [36,38,39]. In our study, inclusion into multivariate analysis of the maternal education and employment status, family economic situation (Model 2), and additionally macroregion according to EU-28 GDP average (Model 3) has not altered the general findings on the association between breastfeeding and later vegetable intake and on the lack of association between breastfeeding and later fruit intake among children. 

It is worth worrying about the fact that only 11% of children in our study consumed vegetables and fruit at least five times a day. Five servings a day of vegetables and fruit (at least 400 g) per person is recommended by the 2002 Joint WHO/FAO Expert Consultation on Diet, Nutrition, and the Prevention of Chronic Diseases [40]. Increasing fruit and vegetable intake is one of the key recommendations of the Global Strategy on Diet, Physical Activity, and Health and is stated as an important approach to prevent childhood obesity [41]. Polish Dietary Guidelines for children aged 4–18 years recommend to eat a variety of vegetables and fruit as often and as much as possible [42]. Diets rich in fruit and vegetables, not only throughout the whole life but specifically during childhood, have been related to lower risks of developing chronic diseases such as diabetes, cancer, and cardiovascular illness, as well as obesity [43]. Those protective effects may result from the presence and combination of many constituents, like antioxidants, fiber, phytoestrogens, isoflavones, coumarins, glucosinolates, and antiinflammatory agents [44,45,46,47].

### Strengths and Limitations

The main strengths of this study were the prospective design and the relatively large representative study population, which allows for high confidence in the study results. It was the national population-based sample, with a good selection of the study group, mother-child pairs, carried out by professional interviewers in the respondents’ homes. Moreover, children’s responses were verified by parents. Additionally, we were able to control for several confounding factors, which is a good basis for the creation of generalizations.

Nevertheless, the current study had some limitations. Firstly, we only examined the frequency of fruit and vegetable consumption, without specifying the size of consumed portions. We are aware of the fact that the frequency of food consumption does not reflect the actual intake of food consumed in an accurate manner. Therefore, we could only test the association between breastfeeding and “almost everyday consumption” of fruit and vegetables in meals. However, other authors also used this method of collecting data [13,14]. Secondly, we lacked information on the maternal diet, and specifically the consumption of vegetable and fruit, which is also a predictor of children’s fruit and vegetable intake [34,35,37,39]. Finally, anthropometric data (height, weight of children) were proxy-reported by the mother; therefore, these data were included only as a confounder factor in logistic regression analysis.

## 5. Conclusions

In conclusion, a longer breastfeeding duration was associated with a higher frequency of vegetable consumption for breakfast and dinner but not with the total daily number of times of vegetable and fruit intake among Polish children aged 7–12 years. Those associations were gender-specific (stronger among girls) and were not diminished by socio-demographic factors. Furthermore, the association between breastfeeding and later fruit consumption was not detected. 

Our study demonstrates some interesting findings that can be directed to formulate relevant, effective intervention programs and policies targeting higher intake of vegetables (and fruit) among children and adolescents. Due to adventitious properties, vegetables and fruit protect against a wide range of illnesses in childhood and adulthood. Moreover, the obtained data could be useful for comparing the results with those from other countries. Still, further studies are also recommended in order to confirm these observations in other European or non-European countries. 

## Figures and Tables

**Figure 1 ijerph-17-03755-f001:**
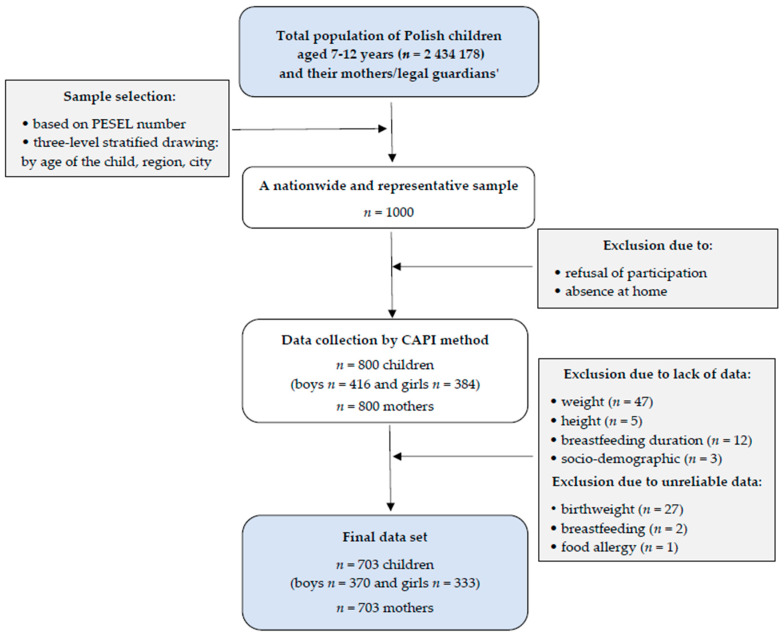
Sample collection chart. PESEL number—number of Universal Electronic System for Registration of the Population.

**Table 1 ijerph-17-03755-t001:** Characteristics of the study population.

Variables		7–9 Years			10–12 Years			*p*-Value ^1,2^ 7–9 vs. 10–12 Years		
	Total Sample 100% (*n* = 703)	Total 53.1% (*n* = 373)	Girls 48.5% (*n* = 181)	Boys 51.5% (*n* = 192)	Total 56.9% (*n* = 330)	Girls 46.1% (*n* = 152)	Boys 53.9% (*n* = 178)	Total	Girls	Boys
**Breastfeeding duration:**								ns	ns	ns
- no	14.5 (102)	13.4 (50)	14.4 (26)	12.5 (24)	15.8 (52)	14.5 (22)	16.9 (30)			
- 0.2–3 months	23.8 (167)	22.3 (83)	21.0 (38)	23.4 (45)	25.5 (84)	24.3 (37)	26.4 (47)			
- 4–6 months	24.5 (172)	28.2 (105)	26.0 (47)	30.2 (58)	20.3 (67)	21.7 (33)	19.1 (34)			
- 7–12 months	24.7 (174)	25.5 (95)	27.1 (49)	24.0 (46)	23.9 (79)	27.0 (41)	21.3 (38)			
- >12 months	12.5 (88)	10.7 (40)	11.6 (21)	9.9 (19)	14.5 (48)	12.5 (19)	16.3 (29)			
**Children BMI centile:**								ns	ns	ns
Median (25th; 75th centile)	60 (32; 82)	60 (31; 84)	58 (27; 84)	63 (34; 84)	60 (35; 78)	59 (28; 81)	61 (40; 76)			
Mean ± SD	55.8 ± 29.7	56.1 ± 31.0	54.6 ± 32.0	57.6 ± 30.1	55.5 ± 28.1	54.7 ± 29.8	56.2 ± 26.7			
Min - Max	0.1 – 99.9	0.1 – 99.0	0.1 – 99.0	0.1 – 99.0	0.1 – 99.9	0.1 – 99.9	1.0 – 98.0			
**Children BMI category:**								ns	ns	ns
- underweight (< 5centile)	5.5 (39)	5.6 (21)	7.2 (13)	4.2 (8)	5.5 (18)	5.9 (9)	5.1 (9)			
- normal (5–84 centile)	74.5 (524)	71.3 (266)	68.0 (123)	74.5 (143)	78.2 (258)	74.3 (113)	81.5 (145)			
- overweight (≥ 85 centile)	12.2(86)	12.9 (48)	14.4 (26)	11.5 (22)	11.5 (38)	13.8 (21)	9.6 (17)			
- obese (≥ 95 centile)	7.7 (54)	10.2 (38)	10.5 (19)	9.9 (19)	4.8 (16)	5.9 (9)	3.9 (7)			
**Maternal education:**								ns	ns	ns
- primary	5.7 (40)	5.9 (22)	4.4 (8)	7.3 (14)	5.5 (18)	6.6 (10)	4.5 (8)			
- vocational	20.3 (143)	19.8 (74)	17.7 (32)	21.9 (42)	20.9 (69)	19.7 (30)	21.9 (39)			
- high school	48.3 (339)	50.1 (187)	51.4 (93)	49.0 (94)	46.1 (152)	46.7 (71)	45.5 (81)			
- university	25.7 (181)	24.1 (90)	26.5 (48)	21.9 (42)	27.6 (91)	27.0 (41)	28.1 (50)			
**Maternal employment status:**								ns	ns	ns
- unemployed	24.2 (170)	28.2 (105)	28.2 (51)	28.1 (54)	19.7 (65)	17.1 (26)	21.9 (39)			
- unearned income	6.1 (43)	5.6 (21)	7.7 (14)	3.6 (7)	5.2 (17)	5.9 (9)	4.5 (8)			
- part-time job	5.4 (38)	60.6 (226)	58.6 (106)	62.5 (120)	68.5 (226)	70.4 (107)	66.9 (119)			
- full-time job	64.3 (452)	5.6 (21)	5.5 (10)	5.7 (11)	6.7 (22)	6.6 (10)	6.7 (12)			
**Number of children:**								ns	ns	ns
- 1	35.3 (248)	35.7 (133)	38.1 (69)	33.3 (64)	34.8 (115)	32.9 (50)	36.5 (65)			
- 2	47.7 (335)	48.3 (180)	46.4 (84)	50.0 (96)	47.0 (155)	48.7 (74)	45.5 (81)			
- ≥ 3	17.0 (120)	16.1 (60)	15.5 (28)	16.7 (32)	18.2 (60)	18.4 (28)	18.0 (32)			
**Self-declared economic situation of household:**								ns	ns	ns
- we live thriftily or poorly	91.3 (642)	91.4 (341)	92.3 (167)	90.6 (174)	91.2 (301)	90.1 (137)	92.1 (164)			
- we live very well	8.7 (61)	8.6 (32)	7.7 (14)	9.4 (18)	8.8 (29)	9.9 (15)	7.9 (14)			
**Place of residence:**								ns	ns	ns
- rural	44.0 (309)	44.5 (166)	44.2 (80)	44.8 (86)	43.3 (143)	42.7 (76)	44.1 (67)			
- urban	56.0 (394)	55.5 (207)	55.8 (101)	55.2 9106)	56.7 (187)	55.9 (85)	57.3 (102)			
**Macroeconomic region of Poland:**								ns	ns	ns
- GDP 47–50%	22.5 (158)	22.0 (82)	23.2 (42)	20.8 (40)	23.0 (76)	20.4 (31)	25.3 (45)			
- GDP 51–100%	61.9 (435)	62.7 (234)	59.7 (108)	65.6 (126)	60.9 (201)	65.1 (99)	57.3 (102)			
- GDP 101–110%	15.6 (110)	15.3 (57)	17.1 (31)	13.5 (26)	16.1 (53)	14.5 (22)	17.4 (31)			

BMI, Body Mass Index centile and categories [18]; GDP, Gross Domestic Product; SD, standard deviation; ^1^ Chi-square Pearson test for all categorical variables; ^2^ U Mann-Whitney test for continuous variable (BMI centile); ns, non-significant.

**Table 2 ijerph-17-03755-t002:** Consumption of meals, vegetables, and fruit among children.

Variables		7–9 Years			10–12 Years			*p*-Value ^1^ 7–9 vs. 10–12 Years		
	Total Sample 100% (*n* = 703)	Total 53.1% (*n* = 373)	Girls 48.5% (*n* = 181)	Boys 51.5% (*n* = 192)	Total 56.9% (*n* = 330)	Girls 46.1% (*n* = 152)	Boys 53.9% (*n* = 178)	Total	Girls	Boys
**Breakfast frequency:**								ns	ns	ns
- never	0.4 (3)	0.8 (3)	1.1 (2)	0.5 (1)	-	-	-			
- rare	3.0 (21)	2.9 (11)	2.8 (5)	3.1 (6)	3.0	2.6 (4)	3.4 (6)			
- frequent	5.4 (38)	4.6 (17)	5.5 (10)	3.6 (7)	6.4 (21)	4.6 (7)	7.9 (14)			
- always	91.2 (641)	91.7 (342)	90.6 (164)	92.7 (178)	90.6 (299)	92.8 (141)	88.8 (158)			
**Second breakfast frequency:**								ns	ns	ns
- never	4.0 (28)	2.9 (11)	2.8 (5)	3.1 (6)	5.2 (17)	4.6 (7)	5.6 (10)			
- rare	18.3 (129)	17.7 (66)	16.0 (29)	19.3 (37)	19.1 (63)	23.7 (36)	15.2 (27)			
- frequent	36.0 (253)	34.4 (128)	37.0 (67)	31.8 (61)	37.9 (125)	35.5 (54)	39.9 (71)			
- always	41.7 (293)	45.0 (168)	44.2 (80)	45.8 (88)	37.9 (125)	36.2 (55)	39.3 (70)			
**Dinner frequency:**								ns	ns	ns
- never	0.1 (1)	0.3 (1)	0.6 (1)	-	-	-	-			
- rare	0.4 (3)	0.8 (3)	1.1 (2)	0.5 (1)	-	-	-			
- frequent	4.4 (31)	3.5 (13)	3.3 (6)	3.6 (7)	5.5 (18)	6.6 (10)	4.5 (8)			
- always	95.1 (668)	95.4 (356)	95.0 (172)	95.8 (184)	94.5 (312)	93.4 (142)	95.5 (170)			
**Supper frequency:**								ns	ns	ns
- never	0.6 (4)	0.3 (1)	-	0.5 (1)	0.9 (3)	1.3 (2)	0.6 (1)			
- rare	2.4 (17)	2.1 (8)	3.3 (6)	1.0 (2)	2.7 (9)	3.9 (6)	1.7 (3)			
- frequent	7.0 (49)	7.8 (29)	7.2 (13)	8.3 (16)	6.1 (20)	9.2 (14)	3.4 (6)			
- always	90.0 (633)	89.8 (335)	89.5 (162)	90.1 (173)	90.3 (298)	85.5 (130)	94.4 (168)			
**Consumed for breakfast:**										
- vegetables	23.9 (168)	22.0 (82)	21.0 (38)	22.9 (44)	26.1 (86)	27.6 (42)	24.7 (44)	ns	ns	ns
- fruit	13.9 (98)	13.4 (50)	14.9 (27)	12.0 (23)	14.5 (48)	14.5 (22)	14.6 (26)	ns	ns	ns
**Consumed for second breakfast:**										
- vegetables	18.6 (131)	15.8 (59)	18.8 (34)	13.0 (25)	21.8 (72)	23.7 (36)	20.2 (36)	0.041	ns	ns
- fruit	46.1 (324)	45.3 (169)	47.0 (85)	43.8 (84)	47.0 (155)	50.7 (77)	43.8 (78)	ns	ns	ns
**Consumed for dinner:**										
- vegetables	47.4 (333)	48.5 (181)	49.7 (90)	47.4 (91)	46.1 (152)	50.0 (76)	42.7 (76)	ns	ns	ns
- fruit	7.7 (54)	9.4 (35)	11.0 (20)	7.8 (15)	5.8 (19)	5.9 (9)	5.6 (10)	ns	ns	ns
**Consumed for supper:**										
- vegetables	26.7 (188)	28.7 (107)	30.9 (56)	26.6 (51)	24.5 (81)	24.3 (37)	24.7 (44)	ns	ns	ns
- fruit	12.9 (91)	14.5 (54)	13.8 (25)	15.1 (29)	11.2 (37)	14.5 (22)	8.4 (15)	ns	ns	0.047
**Consumed between meals:**										
- vegetables	4.0 (28)	4.0 (15)	4.4 (8)	3.6 (7)	3.9 (13)	3.3 (5)	4.5 (8)	ns	ns	ns
- fruit	59.7 (420)	58.7 (219)	63.5 (115)	54.2 (104)	60.9 (201)	68.4 (104) **	54.5 (97) **	ns	ns	ns
**Vegetables and fruit (times per day):**										
- 0	7.1 (50)	7.8 (29)	6.6 (12)	8.9 (17)	6.4 (21)	4.6 (7)	7.9 (14)	ns	ns	ns
- 1	15.6 (110)	15.3 (57)	12.7 (23)	17.7 (34)	16.1 (53)	15.1 (23)	16.9 (30)			
- 2	25.0 (176)	23.9 (89)	21.5 (39)	26.0 (50)	26.4 (87)	19.7 (30)	32.0 (57)			
- 3	28.2 (198)	29.8 (111)	30.9 (56)	28.6 (55)	26.4 (87)	32.2 (49)	21.3 (38)			
- 4	12.9 (91)	12.9 (48)	17.7 (32)	8.3 (16)	13.0 (43)	13.8 (21)	12.4 (22)			
- ≥5	11.1 (78)	10.5 (39)	10.5 (19)	10.4 (20)	11.8 (39)	14.5 (22)	9.6 (17)			

SD, standard deviation. ^1^ Chi-square Pearson test for all categorical variables; ns, non-significant. ** significant difference between girls and boys within age group at *p* ≤ 0.01.

**Table 3 ijerph-17-03755-t003:** The associations between breastfeeding duration and vegetable and fruit consumption for breakfast.

Group	Breastfeeding Duration	Vegetable Consumption			Fruit Consumption		
		Model 1 ^1^ OR ^4^ (95% CI ^5^)	Model 2 ^2^ aOR ^6^ (95% CI)	Model 3 ^3^ aOR (95% CI)	Model 1 ^1^ OR ^4^ (95% CI ^5^)	Model 2 ^2^ aOR ^6^ (95% CI)	Model 3 ^3^ aOR (95% CI)
**Total**	No breastfeeding		Ref. 1			Ref. 1	
	0.2–3 months	2.68 (1.34–5.37) **	2.71 (1.34–5.15) **	2.72 (1.34–5.50) **	0.71 (0.33–1.52)	0.75 (0.35–1.61)	0.75 (0.35–1.61)
	4–6 months	3.92 (1.98–7.73) ***	3.85 (1.93–7.40) ***	3.80 (1.90–7.59) ***	1.27 (0.64–2.54)	1.27 (0.63–2.56)	1.25 (0.62–2.53)
	7–12 months	2.03 (1.00–4.09) *	1.88 (0.92–3.61)	1.93 (0.94–3.96)	1.26 (0.63–2.51)	1.14 (0.56–2.32)	1.18 (0.58–2.41)
	>12 months	1.67 (0.74–3.75)	1.70 (0.75–3.85)	1.74 (0.77–3.95)	0.97 (0.29–1.75)	0.97 (0.29–1.77)	0.98 (0.29–1.80)
**Girls**	No breastfeeding		Ref. 1			Ref. 1	
	0.2–3 months	4.28 (1.37–13.39) **	4.04 (1.26–13.00) *	4.09 (1.26–13.22) *	0.38 (0.12–1.14)	0.41 (0.13–1.26)	0.41 (0.13–1.29)
	4–6 months	5.30 (1.72–16.32) **	4.77 (1.50–15.23) **	4.60 (1.43–14.75) **	0.84 (0.33–2.15)	0.90 (0.34–2.37)	0.87 (0.32–2.33)
	7–12 months	3.35 (1.08–10.41) *	2.87 (0.89–9.23)	2.94 (0.91–9.49)	0.94 (0.38–2.31)	0.89 (0.35–2.26)	0.92 (0.36–2.37)
	>12 months	2.75 (0.76–9.93)	2.97 (0.80–11.01)	3.00 (0.81–11.17)	0.62 (0.19–2.02)	0.59 (0.18–1.97)	0.60 (0.18–2.04)
**Boys**	No breastfeeding		Ref. 1			Ref. 1	
	0.2–3 months	1.92 (0.79–4.65)	2.01 (0.82–4.91)	1.94 (0.79–4.78)	1.33 (0.62–4.14)	1.38 (0.62–4.14)	1.34 (0.43–4.16)
	4–6 months	3.22 (1.36–7.63) **	3.26 (1.36–7.80) **	3.17 (1.32–7.63) **	2.06 (1.04–6.72)	1.97 (1.04–6.72)	1.92 (0.65–4.71)
	7–12 months	1.35 (0.54–3.42)	1.31 (0.51–3.37)	1.30 (0.50–3.35)	1.79 (0.39–2.93)	1.71 (0.39–2.93)	1.74 (0.56–5.35)
	>12 months	1.15 (0.40–3.34)	1.17 (0.40–3.44)	1.19 (0.40–3.51)	1.48 (0.22–3.53)	1.49 (0.23–3.79)	1.39 (0.23–3.78)

^1^ Model 1, unadjusted model; ^2^ Model 2, model adjusted for children age, BMI centile (and gender in total group), maternal education, employment status, and family economic situation; ^3^ Model 3, Model 2 adjusted for macroregion according to EU–28 GDP average; ^4^ OR, odds ratio; ^5^ CI, confidence interval; ^6^ aOR, adjusted odds ratio; * *p* ≤ 0.05; ** *p* ≤ 0.01; *** *p* ≤ 0.001.

**Table 4 ijerph-17-03755-t004:** The associations between breastfeeding duration and vegetable and fruit consumption for dinner.

Group	Breastfeeding Duration	Vegetable Consumption			Fruit Consumption		
		Model 1 ^1^ OR ^4^ (95% CI ^5^)	Model 2 ^2^ aOR ^6^ (95% CI)	Model 3 ^3^ aOR (95% CI)	Model 1 ^1^ OR ^4^ (95% CI ^5^)	Model 2 ^2^ aOR ^6^ (95% CI)	Model 3 ^3^ aOR (95% CI)
**Total**	No breastfeeding		Ref. 1			Ref. 1	
	0.2–3 months	1.43 (0.86–2.38)	1.45 (0.87–2.41)	1.48 (0.88–2.46)	0.96 (0.36–2.55)	1.01 (0.37–2.71)	1.00 (0.37–2.70)
	4–6 months	1.45 (0.90–2.47)	1.46 (0.88–2.42)	1.50 (0.90–2.50)	1.11 (0.43–2.88)	1.16 (0.44–3.04)	1.16 (0.44–3.05)
	7–12 months	1.97 (1.19–3.25) **	1.83 (1.10–3.05) *	1.83 (1.09–3.05) *	1.47 (0.59–3.67)	1.38 (0.54–3.50)	1.37 (0.54–3.49)
	>12 months	2.31 (1.29–4.14) **	2.38(1.32–4.28) **	2.36 (1.30–4.26) **	0.99 (0.32–3.07)	0.99 (0.32–3.08)	0.99 (0.32–3.09)
**Girls**	No breastfeeding		Ref. 1			Ref. 1	
	0.2–3 months	1.68 (0.80–3.55)	1.88 (0.87–4.04)	1.89 (0.87–4.12)	0.79 (0.20–3.09)	0.81 (0.20–3.27)	0.80 (0.20–3.25)
	4–6 months	1.57 (0.75–3.28)	1.75 (0.82–3.76)	1.81 (0.83–3.92)	1.05 (0.29–3.81)	1.05 (0.28–3.95)	1.07 (0.28–4.05)
	7–12 months	2.50 (1.21–5.15) **	2.66 (1.26–5.64) **	2.61 (1.22–5.58) **	1.07 (0.31–3.76)	1.05 (0.29–3.81)	1.04 (0.29–3.80)
	>12 months	2.74 (1.15–6.50) *	3.00 (1.24–7.24) *	3.04 (1.24–7.46) **	1.57 (0.39–6.29)	1.60 (0.39–6.56)	1.59 (0.39–6.51)
**Boys**	No breastfeeding		Ref. 1			Ref. 1	
	0.2–3 months	1.25 (0.63–2.49)	1.24 (0.61–2.49)	1.28 (0.63–2.59)	1.19 (0.28–4.95)	1.27 (0.30–5.38)	1.26 (0.30–5.36)
	4–6 months	1.43 (0.72–2.84)	1.40 (0.70–2.83)	1.45 (0.72–2.95)	1.19 (0.28–4.95)	1.23 (0.29–5.26)	1.23 (0.29–5.24)
	7–12 months	1.55 (0.77–3.11)	1.42 (0.69–2.90)	1.42 (0.69–2.94)	2.04 (0.53–7.90)	2.05 (0.51–8.03)	2.05 (0.51–8.24)
	>12 months	2.01 (0.91–4.44)	2.18 (0.97–4.89) *	2.20 (1.01–4.95) *	1.36 (0.04–3.60)	1.37 (0.04–3.56)	1.37 (0.04–3.71)

^1^ Model 1, unadjusted model; ^2^ Model 2, model adjusted for children age, BMI centile (and gender in total group), maternal education, employment status, and family economic situation; ^3^ Model 3, Model 2 adjusted for macroregion according to EU–28 GDP average; ^4^ OR, odds ratio; ^5^ CI, confidence interval; ^6^ aOR, adjusted odds ratio; * *p* ≤ 0.05; ** *p* ≤ 0.01;.

**Table 5 ijerph-17-03755-t005:** The associations between breastfeeding duration and vegetable and fruit consumption for supper.

Group	Breastfeeding Duration	Vegetable Consumption			Fruit Consumption		
		Model 1 ^1^ OR ^4^ (95% CI ^5^)	Model 2 ^2^ aOR ^6^ (95% CI)	Model 3 ^3^ aOR (95% CI)	Model 1 ^1^ OR ^4^ (95% CI ^5^)	Model 2 ^2^ aOR ^6^ (95% CI)	Model 3 ^3^ aOR (95% CI)
**Total**	No breastfeeding		Ref. 1			Ref. 1	
	0.2–3 months	1.13 (0.65–1.95)	1.14 (0.66–1.97)	1.13 (0.65–1.96)	0.77 (0.38–1.56)	0.80 (0.39–1.63)	0.80 (0.39–1.63)
	4–6 months	0.96 (0.56–1.67)	1.03 (0.54–1.63)	1.01 (0.52–1.59)	0.59 (0.28–1.23)	0.55 (0.26–1.16)	0.54 (0.26–1.14)
	7–12 months	1.07 (0.62–1.84)	1.03 (0.59–1.80)	1.06 (0.60–1.85)	0.86 (0.43–1.71)	0.71 (0.35–1.44)	0.73 (0.36–1.47)
	>12 months	0.50 (0.24–1.03)	0.51 (0.25–1.04)	0.51 (0.25–1.06)	0.93 (0.42–2.06)	0.92 (0.41–2.05)	0.94 (0.42–2.11)
**Girls**	No breastfeeding		Ref. 1			Ref. 1	
	0.2–3 months	1.76 (0.35–2.64)	1.75 (0.34–2.74)	1.76 (0.34–1.70)	1.01 (0.36–2.81)	1.04 (0.36–2.97)	1.04 (0.36–2.99)
	4–6 months	1.69 (0.32–2.56)	1.63 (0.28–2.40)	1.63 (0.28–1.42)	0.56 (0.18–1.71)	0.51 (0.16–1.62)	0.49 (0.15–1.55)
	7–12 months	1.74 (0.35–2.22)	1.65 (0.30–2.41)	1.69 (0.32–2.49)	1.08 (0.40–2.89)	1.08 (0.35–2.69)	1.00 (0.36–2.76)
	>12 months	1.32 (1.08–1.92) *	1.30 (1.11–1.87) *	1.31 (1.10–1.89) *	1.46 (0.48–4.46)	1.50 (0.48–4.64)	1.54 (0.49–4.79)
**Boys**	No breastfeeding		Ref. 1			Ref. 1	
	0.2–3 months	1.71 (0.77–3.80)	1.77 (0.79–3.96)	1.71 (0.76–3.85)	0.61 (0.23–1.61)	0.61 (0.22–1.65)	0.58 (0.21–1.60)
	4–6 months	1.38 (0.61–3.10)	1.35 (0.60–3.07)	1.30 (0.57–2.98)	0.61 (0.23–1.61)	0.53 (0.19–1.46)	0.51 (0.18–1.40)
	7–12 months	1.56 (0.69–3.53)	1.76 (0.76–4.05)	1.78 (0.76–4.12)	0.68 (0.26–1.79)	0.51 (0.18–1.44)	0.50 (0.18–1.41)
	>12 months	1.78 (0.29–2.14)	1.83 (0.30–2.29)	1.83 (0.30–2.31)	0.58 (0.18–1.87)	0.58 (0.17–1.94)	0.56 (0.17–1.91)

^1^ Model 1, unadjusted model; ^2^ Model 2, model adjusted for children age, BMI centile (and gender in total group), maternal education, employment status, and family economic situation; ^3^ Model 3, Model 2 adjusted for macroregion according to EU–28 GDP average; ^4^ OR, odds ratio; ^5^ CI, confidence interval; ^6^ aOR, adjusted odds ratio; * *p* ≤ 0.05;.

**Table 6 ijerph-17-03755-t006:** The associations between breastfeeding duration and total daily number of servings of vegetables and fruit (0 or ≥ 5 times per day).

Group	Breastfeeding Duration	Vegetable and Fruit Consumption 0 x per Day			Vegetable and Fruit Consumption ≥ 5 x per Day		
		Model 1 ^1^ OR ^4^ (95% CI ^5^)	Model 2 ^2^ aOR ^6^ (95% CI)	Model 3 ^3^ aOR (95% CI)	Model 1 ^1^ OR ^4^ (95% CI ^5^)	Model 2 ^2^ aOR ^6^ (95% CI)	Model 3 ^3^ aOR (95% CI)
**Total**	No breastfeeding		Ref. 1			Ref. 1	
	0.2–3 months	0.40 (0.15–1.09)	0.39 (0.14–1.06)	0.38 (0.14–1.04)	1.74 (0.71–4.30)	1.92 (0.76–4.81)	1.93 (0.77–4.88)
	4–6 months	0.15 (0.20–1.30)	0.51 (0.20–1.31)	0.50 (0.19–1.29)	1.99 (0.82–4.84)	2.06 (0.83–5.10)	2.03 (0.81–5.05)
	7–12 months	0.56 (0.23–1.40)	0.61 (0.24–1.54)	0.59 (0.23–1.51)	1.86 (0.76–4.55)	1.57 (0.63–3.91)	1.64 (0.66–4.10)
	>12 months	0.74 (0.14–1.73)	0.71 (0.11–1.71)	0.75 (0.72–1.41)	1.55 (0.55–4.34)	1.60 (0.56–4.57)	1.65 (0.58–4.74)
**Girls**	No breastfeeding		Ref. 1			Ref. 1	
	0.2–3 months	0.46 (0.10–2.14)	0.42 (0.09–2.01)	0.39 (0.08–1.89)	1.50 (0.43–5.17)	2.02 (0.55–3.40)	2.08 (0.56–3.71)
	4–6 months	0.43 (0.09–2.00)	0.43 (0.09–2.07)	0.45 (0.09–2.21)	1.57 (0.46–5.32)	1.91 (0.53–3.91)	1.86 (0.51–3.82)
	7–12 months	0.51 (0.12–2.14)	0.58 (0.13–2.36)	0.59 (0.13–2.59)	1.86 (0.57–5.05)	1.89 (0.55–3.52)	1.98 (0.57–3.84)
	>12 months	0.57 (0.13–2.22)	0.51 (0.21–2.24)	0.51 (0.36–2.32)	1.57 (0.39–5.29)	2.02 (0.47–3.42)	2.09 (0.49–3.88)
**Boys**	No breastfeeding		Ref. 1			Ref. 1	
	0.2–3 months	0.36 (0.10–1.35)	0.36 (0.10–1.35)	0.36 (0.09–1.34)	2.07 (0.54–7.89)	2.12 (0.54–8.24)	2.08 (0.53–8.22)
	4–6 months	0.56 (0.17–1.83)	0.58 (0.18–1.92)	0.57 (0.17–1.90)	2.55 (0.69–9.48)	2.53 (0.66–9.66)	2.57 (0.66–9.94)
	7–12 months	0.62 (0.19–2.02)	0.61 (0.18–2.05)	0.61 (0.18–2.08)	1.79 (0.45–7.07)	1.47 (0.36–5.97)	1.57 (0.38–6.49)
	>12 months	0.85 (0.60–5.64)	0.80 (0.58–5.58)	0.81 (0.58–5.64)	1.55 (0.33–7.28)	1.60 (0.33–7.75)	1.45 (0.30–7.12)

^1^ Model 1, unadjusted model; ^2^ Model 2, model adjusted for children age, BMI centile (and gender in total group), maternal education, employment status, and family economic situation; ^3^ Model 3, Model 2 adjusted for macroregion according to EU–28 GDP average PKB region; ^4^ OR, odds ratio; ^5^ CI, confidence interval; ^6^ aOR, adjusted odds ratio.

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
