# Peer review of "Analysis of Association between Breastfeeding and Vegetable or Fruit Intake in Later Childhood in a Population-Based Observational Study"

_ijerph, 2020, doi:10.3390/ijerph17113755_

Round 1
Reviewer 1 Report
Sample size: reasonable sample size (mother and child pairs)
Samples age group: Literature reviews discuss children under 7 or up to age 7. In this study age group selected is 7-12 (because they could ask questions directly to children?) – This is appropriate.
Anthropometry: Actual weight and height not measured – addressed in the limitations
Vegetables – Include potatoes? What types of vegetables and fruits?
Mother’s education: ‘high’ – meaning? (high school?)
Strengths and limitations – well discussed
Minor English corrections:
Line 253 none – change to ‘no’
Line 275 can – change to ‘may’ seems to read better
Line 297 till – change to ‘to’
306 references – [ missing
Recommendation
Make the few minor corrections mentioned above.
The paper is an interesting addition to the literature and worthy of special publicity.
Author Response
Dear Reviewer,
We greatly appreciate your work, helping to strengthen our manuscript. Below (in Attachment) we have included all the Reviewer’s comments along with our answers. Changes made in the manuscript text have been highlighted in yellow for greater readability.
We hope to have addressed all the Reviewer’s points satisfactorily.
Looking forward to hearing from you,
Yours sincerely,
Jadwiga Hamułka

Reviewer 2 Report
The objective of this research was to clarify the associations between breastfeeding duration in infancy and fruit and vegetable intake among children 7-12 years old. The introduction clearly explains the need to understand various confounding factors that may influence the association between between breastfeeding. The study design addresses these questions. However, the presentation of the results in the title, abstract, results, and discuss are misleading of the true findings of the trial. Regardless of the model used to analyse the data, there was no association between breastfeeding duration and fruit or vegetable consumption (Supplemental Table 3). The results presented focus on the association to vegetable intake at certain meal times, but the mechanism for this observed association is unclear. The clinically relevant outcome is total servings of fruit and vegetables per day, which there is a null association. These null findings should be the focus of the paper, and not reserved for the supplement.
Additional revisions are attached.

Author Response

(The authors gave the same response as above.)

Reviewer 3 Report
The research has been conducted very well, both in terms of methodology as well as statistics, on a representative, considerably large group of Polish children. However analysis of the connection between the breastfeeding and eating vegetables and fruit by children is a bit surprising.
What I would like to particularly emphasize is the manuscript, written in a clear manner, easy to read and understand for the reader. Appropriate bibliography have been used in the text of article.
However, I would like to point out some minor issues regarding the content:
- in point 2.3. (line 126), the authors mentioned that the children were asked about their consumption of “lunch” and “dinner”. Meanwhile, the following part of the article analyzes only the consumption of “dinner” without any reference to “lunch” (neither in lines 176, 196, nor in the tables). What is, therefore, the reason for differentiating between "lunch" and "dinner" in the research methodology ? Why it wasn't applied in results of the study ?
- in the research methodology, data regarding the children's height and weight are compiled, which are later converted to BMI. However, the research results do not indicate any potential relationship between the children's BMI and consumption of fruit and vegetables. There is no mention about it in the discussion. So what was the purpose for collecting this data?
- in point 2.3. (line 130), instead of the generalization "several times a day", I would suggest to specify exactly how often vegetables and fruit were eaten, as in table 2, i.e. twice a day, three times a day, etc.
- the period of breastfeeding (0-3 months) is a little unclear. Zero could generally be understood as lack of breastfeeding. In line 134, the authors mention that mothers were asked about breastfeeding time in months or weeks. Perhaps it would be better to indicate the period as (...weeks – 3 months) or (1 – 3 months) ?
Technical remarks:
- in line 202, the authors list the percentage of boys and girls, with the accuracy to the decimal point, whereas earlier in the entire paragraph 3.2. the values ​​are rounded to whole numbers.
- there is quite a serious error in line 220. The OR value is missing, which according to Table 4 equals “2.20”. The “CI” value is also incorrect.
- in line 306, brackets are necessary [35,37,38]
Author Response
Dear Reviewer
We greatly appreciate your work, helping to strengthen our manuscript. Below (in Attachment) we have included all the Reviewer’s comments along with our answers. Changes made in the manuscript text have been highlighted in yellow for greater readability.
We hope to have addressed all the Reviewer’s points satisfactorily.
Looking forward to hearing from you,
Yours sincerely,
Jadwiga Hamułka
